# Sustainability of the Artisanal and Small-Scale Gold Mining in Northeast Antioquia-Colombia

**Óscar Jaime Restrepo Baena \*** and **Luis Enrique Martínez Mendoza \***

School of Mines, Universidad Nacional de Colombia, 050010 Medellín, Colombia
\* Correspondence: ojrestre@unal.edu.co (Ó.J.R.B.); luemartinezme@unal.edu.co (L.E.M.M.)

**Abstract:** The aim of this work is to explain the concepts of sustainability with respect to small artisanal gold mining. For this, a qualitative approach with a descriptive scope was used, for which the bibliographic review technique was conducted. In this sense, articles, theses, books and institutional documents, and any contribution related to the research topic were taken into consideration. Likewise, this documentation contributed to the delimiting aspects that allowed a contrast between the proposed definitions and small artisanal mining in the Northeast Antioquia region in Colombia. Based on the reviewed sources, different needs were recognized in artisanal small-scale gold mining in Northeast Antioquia that still need action. In conclusion, through the exposition of sustainability theories, three common factors were identified within the various positions that were raised—the environmental, economic, and sociocultural dimensions.

**Keywords:** artisanal mining; small-scale gold mining; sustainability in mining; gold mining



## 1. Introduction

For industry, applying the concept of sustainability can be a complex task, owing to the broad scope of the term, the dynamism with which it varies over time, and the influence of the culture of the society. These aspects have given rise to conflicts between the different currents that have adopted the term since it emerged as world public policy in 1992 [1]. In this sense, the economic and environmental spheres are the two dimensions of sustainability that have presented the greatest conflict, even in the decades before the nineties [1].

An example of the above is the definition provided by Giovannoni and Fabietti [2], who ensured that sustainability, from the business point of view, refers to the ability of a corporation to last over time in terms of profitability, productivity, and financial and management performance of the environmental and social assets that make up its capital. In short, business sustainability is the business of staying in business.

Thus, it can be seen that the industrial perspective first welcomes the economic dimension, while the environmental and social aspects remain secondary and move away from the first motivation that promoted sustainability: the capacity of the environment to meet current and future needs.

The following question then arises: what happens when there is no corporation? Or, more importantly, when there are rudimentary or informal industrial processes, are the environmental and social dimensions considered? In this order of ideas, artisanal and small-scale gold mining emerges as the main topic of this research, an activity characterized by its rudimentary processes, its tendency towards informality, and its high presence in rural areas [3]. In this type of mining, a large part of the population lacks the technical knowledge to properly exploit the resources [3].

Regarding the environmental issue, by employing rudimentary processes, artisanal and small-scale gold mining stands out worldwide for its use of mercury for the treatment of gold minerals, whether from alluvial deposits or veins. This results in significant

emissions into the atmosphere and discharge into bodies of water and soil. Precisely, these last two mean the favoring of mercury methylation, which is a process that generates bioavailability in people's diet [4].

On the other hand, in Latin America, small artisanal gold mining is characterized as a subsistence method for many people. This culture has not only produced environmental effects but also favored social problems, ranging from territorial disputes over access to deposits to the forced labor of children [5].

In Colombia, this type of mining has been fundamental throughout its history, as most of its gold production comes from small artisanal mining [6]. However, despite the government's efforts to prohibit the commercialization and use of mercury and to initiate mining formalization processes, there is still much work to be done if it is to enter the path of sustainability, especially in the social and environmental dimensions, since issues such as child labor or the contamination of soils and water bodies with mercury persist.

At the regional level, the Department of Antioquia plays a fundamental role in the country's gold production, producing about 65% of the national production of gold [7]. However, at the same time, this region was classified as the place with the highest air contamination caused by mercury in the world in 2021 because most of its gold is produced through artisanal methods [8].

As for Northeast Antioquia, it is characterized by its historical gold production, dating from colonial times [9]. This region stands out for the presence of both large-scale and small artisanal mining and is made up of the following municipalities: Amalfi, Yalí, Anorí, Cisneros, Yolombó, Vegachí, Santo Domingo, San Roque, Segovia, and Remedios. These last two municipalities have the highest production with 1635.9 and 1207.8 kg, respectively, in the first quarter of 2020 [7]. Likewise, by 2011, there were 75 artisan plants (or "entables") in Segovia and 22 in Remedios, which accounts for the intense artisanal activity present in this region [10].

Although there is a government initiative to strengthen the sustainability of the mining sector in Colombia, which includes the situation of small artisanal gold mining, there is no evidence of an operational plan to contribute to the sustainability of this area [11]. Furthermore, considering that this type of mining is characterized by the use of rudimentary processes, it is very difficult to expect them to take the initiative in the implementation of sustainability plans.

Artisanal and small-scale gold mining has favored the contribution of certain Sustainable Development Goals (SDG) to the detriment of others. For example, in Northeast Antioquia, small artisanal gold mining has contributed to poverty reduction and hunger reduction (Objectives 1 and 2). However, it has also affected health and sustainable water management (Goals 3 and 6) [12]. Therefore, if measures are not taken, this type of mining will not be sustainable since it would affect the interrelation between the SDGs [12].

That said, this research aims to explain the concepts of sustainability and relate them to small artisanal gold mining. As for the scope of this research, is expected that, first, this research will become an antecedent for future projects and initiatives associated with the sustainability of the small-scale Artisanal gold mining in Northeast Antioquia, and thus, will contribute to the phenomenon of study in a context with little inquiry. Second, this article will contribute to the image and acceptance of mining in Colombia through the exposition of concepts that allow actual problems such as the approach with communities to be addressed.

## 2. Methodology

The methodology used is that of a qualitative approach with a descriptive scope [13]. The technique used for the development of the objectives was documentary analysis [14]. For this, articles, theses, books and institutional documents, and any contribution related to the research topic were taken into consideration.

According to Sandoval [14], documentary analysis is a technique that helps to deepen and interpret the reality of documents, and it has five steps: The first step is to identify

the documents regarding the research topic. The second step is to classify the identified documents in an orderly manner, including the necessary bibliographic data. The third step is to select the documents that are better adjusted to the researcher's needs and the research problem. This means that not all of the documents identified in the first step are useful, which is why there is a need to implement a second bibliography selection. The fourth step consists of reading the selected documents strategically. This aims to extract the most valuable aspects (related to the subject of study) of each document. Finally, in step five, the information that has been collected is used to build a synthesis that offers a panorama of the research problem concerning the selected documents.

## 3. Theoretical Framework

### 3.1. Artisanal and Small-Scale Gold Mining in Northeast Antioquia

Northeast Antioquia is a subregion of the Department of Antioquia, located east of the Central Cordillera, southwest of the San Lucas mountain range, and between the Porce, Nechí, Nus, and Alicante rivers [15]. It is bordered to the west by the north of the Department of Antioquia, to the northeast by the Department of Bolívar, to the southeast by the Magdalena Medio, to the south by the East of Antioquia, and to the north by the Bajo Cauca region [16].

This subregion has an area of 8544 km$^2$, which is equivalent to 13.6% of the total area of the department [15]. Administratively, Northeast Antioquia is divided into three subregions: Meseta, made up of Anorí, Amalfi, Yolombó, Yalí, and Vegachí; Minera, which brings together Segovia and Remedios; and El Nus, which is made up of the municipalities of Cisneros, San Roque, and Santo Domingo [15]. In Figure 1 is illustrated the map of the Department of Antioquia with the aforementioned territories.

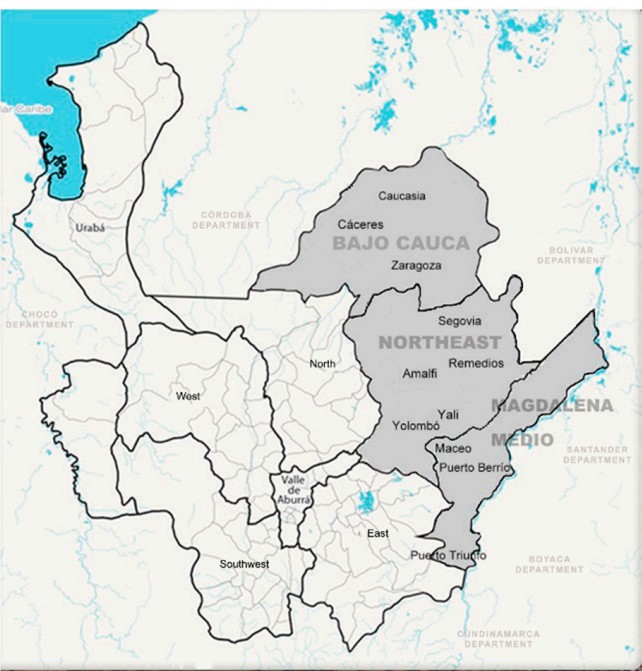

**Figure 1.** Map of the Department of Antioquia. Source: Henao (2015).

The municipalities of Segovia and Remedios stand out as those with the greatest relationship with artisanal and small-scale gold mining, which is evidenced in Table 1, which details the last mining unit census that was conducted. There, it is also observed that underground mining is the predominant type of artisanal mining in this territory.

**Table 1.** Mining units in Northeast Antioquia.

| Municipality | Remedios | Segovia |
|---|---|---|
| Underground exploitations | 109 | 146 |
| Workers, mines and partners | 2150 | 4000 |
| Entables or artisan plants | 22 | 75 |
| Amalgamating mills | 1874 | 526 |
| Entable workers | 38 | 362 |

Source: García and Molina [10].

Although municipalities such as San Roque, Yalí, and Yolombó are not classified within the Northeast Antioquia mining (Minera) subregion, the presence of small artisanal gold mining also stands out.

In 2010, the Department of Antioquia was listed as the place with the highest air contamination caused by mercury in the world [17]. In the same year, most of the "entables" in charge of gold transformation processes were generating significant mercury losses and vapor emissions into the atmosphere [17]. In addition, these were located near schools, shops, and houses and caused a direct impact on the urban population [17].

One of the main sources of emissions from these "entables" was found in the amalgamating mills, which received an excessive amount of mercury, ranging between 100 and 120 g of mercury for every 60 or 70 kg of ore to be transformed [17]. Additionally, the equipment was used at a speed that exceeded the critical speed by 35%, which made the amalgamation process highly inefficient and caused mercury losses of up to 50% (emitted into the atmosphere in the form of steam or discarded into bodies of water with the process residues) [17].

However, after 2010, the University of British Columbia led a process intending to reduce mercury emissions resulting from this mining activity. In this regard, the university-trained plant owners were to change certain processes used in their plants, such as reducing the rotation speed of the mills or the use of activated mercury [18]. With these two new processes, it was possible to reduce the amount of mercury entering the system, which went from 78.1 g on average for each mill to 44.3 g, which represented a reduction of close to 43% [18]. Similarly, it was possible to reduce the amount of mercury lost by each amalgamating mill, which went from 36.1 to 13.5 g and represented an approximate reduction of 63% [18].

Similarly, a second training session was conducted at the Portovelo plant in Ecuador, in which the plant owners were trained to go from amalgamating all of the extracted mineral to implementing more efficient processes for the recovery of gold, such as gravimetric concentrators, flotation, and cyanidation with activated carbon, among others [18].

Another method introduced in Northeast Antioquia after 2010 was retorting, a device that is used to capture the mercury that evaporates during the burning of gold amalgam and prevents the vapors from being emitted into the atmosphere and from being inhaled by people directly [19]. Through a condensation system (which can be as simple as using wet sand), it is possible to collect the liquid mercury for reuse [19].

Despite all of these efforts led by the Government of Antioquia, UNIDO, and the University of British Columbia, recent studies show that the mercury problem resulting from this mining activity continues to be worrisome. In 2019, Sánchez et al. [20] prepared the mercury baseline for the Department of Antioquia. For this, they defined basic sampling units of 600 km$^2$ that intercept the Department of Antioquia in 152 cells, as shown in Figure 2. Then, they took a total of 464 fine active sediment samples from streams to obtain the average geochemical composition of the drainage basins.

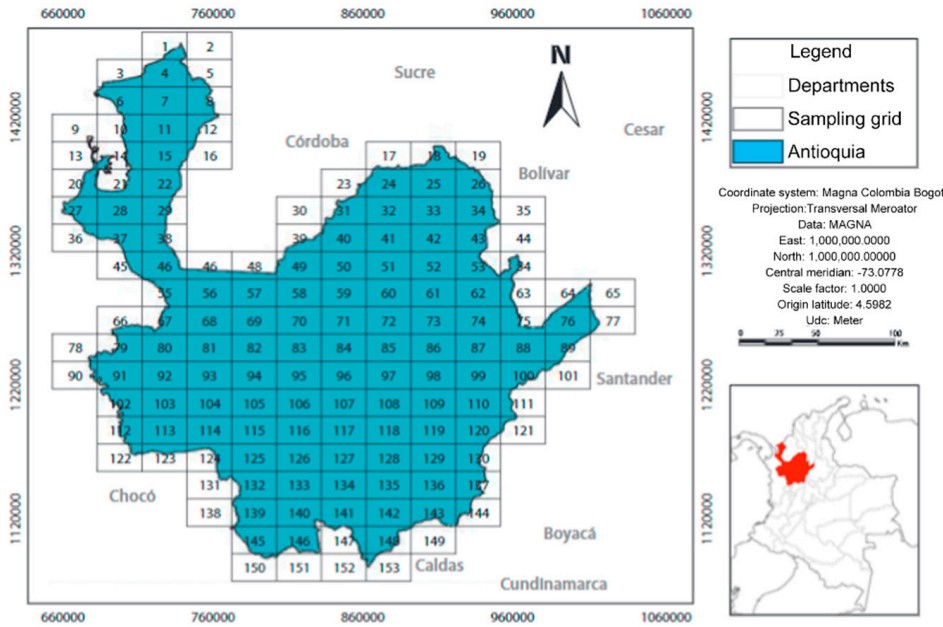

**Figure 2.** Basic sampling units. Source: Sánchez et al. [20].

With these samples, a representative base of 98 panels was formed, which allowed the use of descriptive statistics for data analysis. Through this, a median for the mercury content in the fine active sediments of 67 µg kg$^{-1}$ and a variable range of 11 and 3888 µg kg$^{-1}$ was found. From these results, they defined percentiles that allowed for the classification of the Department of Antioquia according to the concentration of mercury in the active fine sediment current, as shown in Figure 3. Similarly, Table 2 shows the mercury concentration in the active fine sediment current according to the percentiles detected after the analysis of the results.

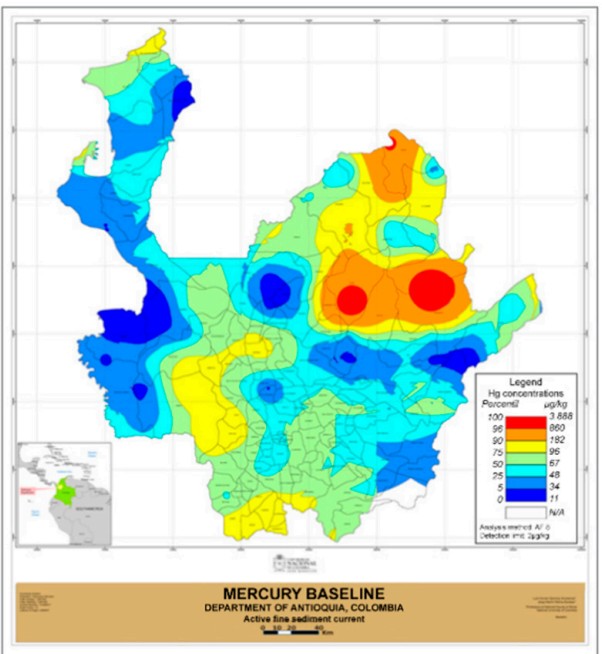

**Figure 3.** Mercury baseline map in the Department of Antioquia. Source: Sánchez et al. [20].

**Table 2.** Mercury distribution percentiles in the active fine sediment current.

| Percentile [%] | Concentration [$\mu g \ kg^{-1}$] |
|:---:|:---:|
| 5 | 3.4 |
| 25 | 48 |
| 50 | 67 |
| 75 | 98 |
| 90 | 182 |
| 98 | 860 |

Source: Sánchez et al. [20].

In Figure 3, it can be observed how the highest values of mercury concentration in the fine active sediments of streams are found in Northeast Antioquia, with minimum values of 182 $\mu g \ kg^{-1}$ and maximum values of 3888 $\mu g \ kg^{-1}$ [20]. These figures reflect the excessive use of mercury for gold amalgamation.

On the other hand, Molina et al. [21] collected samples of breast milk from 50 mothers from Segovia and 17 mothers from Remedios to measure the concentration of mercury that was present in them. For this, each mother was asked to collect 15 milliliters of breast milk in test tubes at any time of the day. After analyzing the samples, it was found that 10.2% of the samples from Segovia exceeded the permissible limit for mercury, while in Remedios, it was 5.6% [21].

In addition to the above, in these municipalities, fish is one of the main sources of protein in the maternal diet, which highlights the transfer of inorganic mercury from the environment to the food consumed and, later, to breast milk [21]. Inorganic mercury is transferred through exposure to high concentrations of mercury vapors, a product of the proximity of homes to gold stores, where amalgam burning is conducted [21].

Amalgamation has traditionally been used to recover fine gold from alluvial mining, from free gold, from milling, and from gravimetric concentrators [22]. However, in Northeast Antioquia, mercury has been used to amalgamate the complete mineral that comes from exploitation work (Whole ore amalgamation) [17]. The complete amalgamation of the mineral consists of putting 100% of the mineral in contact with mercury without previously separating the part that has no economic value [23]. This process is considered highly inefficient, and it is seen as a poor practice for gold recovery [23].

When this process is chosen, mercury use ranges are excessive and can vary as follows:

- High: This range occurs when four parts of mercury are used for every part of recovered gold [23]. Veiga [17] reported that in Northeast Antioquia, there is no mass balance to determine this proportion. However, he proposed an approximate figure of 10:1.
- Very high: This range occurs when twenty parts of mercury are used for each part of extracted gold [23].
- Extreme: This range occurs when up to fifty parts of mercury are used for each part of recovered gold, which makes up a ratio of 50:1 [23].

In addition to the above, when whole ore amalgamation is used, the efficiency of gold recovery is very low, up to 30% in exceptional cases, so much of the gold is lost in the waste or tailings of the process [23]. Veiga [17] reported a 50% gold recovery in the processing plants of Northeast Antioquia, but this figure is based on assumptions (as well as on the radius of the use of mercury).

In another order of ideas, culture has a great influence on how to address sustainability, and depending on how politics work in each country, areas such as the environment involve the most influential people involved in decision-making. For example, in the case of Canada, in the last thirty years, environmental value has gained enough strength to influence decision-making [24]. This is the opposite of the Colombian case, in which sociocultural sustainability may be more oriented towards reducing hunger, poverty, and increasing health conditions, which minimizes the time to reflect on the needs of future generations [24].

As a consequence, in the gold transformation process, Canadian artisanal miners use manual gravimetric methods, such as pan washing, and it can take up to two hours to process gold concentrates [4]. In contrast, in regions where amalgamation is used, such as in Northeast Antioquia, decision-makers are driven by the speed of the process, which can take up to ten minutes, its ease, and its safety (since it does not risk the loss of the finest fractions). Therefore, the two hours that Canadian miners spend generating added value are used by those who prefer amalgamation to continue mining operations [4].

### 3.2. Concepts of Sustainability

Defining sustainability in general can be a complicated task. Indeed, Giovannoni and Fabietti [2] recognized that the ubiquity of the term has caused the arising of different discourses over time, some that are more associated with the concept of social responsibility others that are more attached to the environmental component, and a few more that are linked to the field of economy. This coincides with Lopez [1], who affirms that defining sustainable development and sustainability is a difficult task because of the variety of postures and co-existing definitions in the available literature.

Moreover, Porritt [25] defined sustainability as the capacity of continuity in the long-term future and as the ultimate goal and desired destination of all species. To this conception, we add that of Eggert [26], who proposed that sustainability can be classified into three areas: environmental, economic, and sociocultural:

Eggert [26] also defined that each form of sustainability can be seen as one dimension, which aims to support environmental, economic, or social components. Each of these always gives priority to a specific target (such as the Human Development Index) but at the risk of neglecting the goals of the other components of sustainability.

In 1989 Karl-Henrik Robert launched The Natural Step initiative to provide a simple and clear definition of sustainability [27]. With this, he sought to create a common language that would allow him to work with companies and cities for the planning of this concept.

This initiative raised four principles from which in a sustainable society [27]:

- Nature should not be exposed to systematic increases in the concentration of substances in the earth's crust. For example, rivers should not be exposed to systematic increases in mercury concentration.
- Nature should not be exposed to systematic increases in the concentration of substances produced by society. This means, for example, that society can produce in a more efficient way to reduce waste.
- Nature should not be exposed to systematic increases in physical degradation. This means the efficient management of resources and land, which should always be managed with caution regarding making modifications to nature.
- There should be no structural barriers that affect people's health, influence, competence, or impartiality. This means that products and services must be offered, and different business models and practices must be implemented to ensure that human rights are fulfilled and that communities have living conditions that allow them to satisfy their needs.

### 3.3. Sustainability and Mining

Eggert [26] stated that sustainability in mining does not mean sustaining the life of a mine indefinitely or that of a community. For this author, sustainability is based on four principles:

- Facilitating the creation of mineral wealth: As mineral deposits do not form in short periods, it is first necessary to develop knowledge and study of the occurrence (or probability of occurrence). This is practiced to create mineral wealth that contributes to sustainability [26].
- Ensuring that mineral development is socially and economically efficient: This principle focuses on the maximization of benefits from economic efficiency, that is, a mining project must be very clear about its expenses, incomes, and indirect benefits at the

time of assessing sustainability [26]. Indirect benefits are those that would not occur without the presence of the project in the community, such as local purchases (like protection elements, fuel, food, etc.) or contributions to increase health quality and infrastructure. However, these indirect benefits can also have consequences such as increases in alcoholism, prostitution, environmental damage, among others. Therefore, this principle considers that the project should compensate, for these negative consequences, called indirect costs in the same way [26].

- Distributing surpluses from mining development fairly: Most of the time, mining projects generate surpluses, even after compensating for indirect costs [26]. However, there is an unknown as to how these are distributed fairly, whereupon Eggert mentions three different currents of thought about what is fair. The first, which is Aristotelian, emphasizes proportional distribution; however, this may mean that the profits are distributed among the partners in proportion to their contribution to the surplus. The second current of thought is based on the author Jeremy Bentham [26], who argues that the surplus must be distributed in a way that contributes to the greater good; however, who determines what is the greater good? The company through reinvestment in mineral development? The State through the construction of schools? The third current, which is proposed by John Rawls [26], maintains that this surplus should be destined to contribute to the least benefited groups of society. In this sense, for mining, this approach represents the investment of surpluses in activities that help reduce poverty [26].

- Sustaining the benefits of mining, even after its closure and abandonment: Even though it was mentioned that from Eggert's perspective, sustainability in mining does not consist of indefinitely sustaining a community; the benefits of the operation can be extended permanently [26]. This principle is fundamental for the contribution of sustainability. Thus, its main idea lies in the fact that the company should save and invest part of its income and then periodically reinvest the profits from that investment to sustain spending indefinitely [26].

In addition to the authors who define it, there have also been projects and initiatives to apply the concept of sustainability in mining. The Mining, Minerals, and Sustainable Development [24] centered its concept of sustainability on the welfare of the environment and humans. Thus, it defined sustainability as the parallel care and respect that must exist for the environment and for the people who inhabit it [24].

From this concept, it was proposed that the success of a mining project should be measured by achievement or contribution to the welfare of human beings and the ecosystem together [24]. Therefore, a mine is considered to reduce its potential for sustainability when, as a result of its operations, there is a degradation to the well-being of humans and the environment [24].

The MMSD project assured that it cannot deny the fact that mining activities have impacts that can degrade the environment and the well-being of a community. However, it also assured that a mining project must consider its entire life cycle to make positive contributions to sustainability. In this way, when a project of this type is presented, it can contribute to sustainability as long as it works to strengthen communities and to rehabilitate degraded ecosystems, thus turning mining into a bridge to the future [24].

## 4. Results

With all of these positions exposed, it can be seen that sustainability can be applied in mining despite being an activity related to non-renewable resources, especially if it mainly focuses on the implications it has on communities and the environment [24]. Additionally, from the classification proposed by Eggert [26], it was possible to identify three dimensions: environmental sustainability, economic sustainability, and sociocultural sustainability. Therefore, the results were classified into three points:

### 4.1. Environmental Sustainability

In Northeast Antioquia, work has been done on sustainability without speaking of a specific plan in this regard. However, how has this happened? Taking a look at the principles proposed by The Natural Step [27], it can be seen that the first principle holds that in a sustainable society, nature should not be exposed to systematic increases in the concentration of substances on the earth's crust. Thus, through initiatives such as the one from the University of British Columbia (in conjunction with local governments and UNIDO), in which the concentrations of mercury emitted into the environment have been reduced, it has been possible to make contributions to the sustainability of the artisanal and small-scale gold mining in Northeast Antioquia.

These actions have also contributed to the second principle proposed by The Natural Step [27], which seeks that, in a sustainable society, nature is not exposed to systematic increases in the concentration of substances produced by society. In this sense, by training the plant owners to modify the gold transformation processes, it has been possible to produce more efficiently, and the waste produced by this activity has been reduced. In this way, it has also contributed to the sustainability of artisanal and small-scale gold mining in Northeast Antioquia.

The studies proposed by Molina et al. [21] and Sánchez et al. [20] make it clear that although contributions to the sustainability of artisanal and small-scale gold mining in Northeast Antioquia and the first two principles of The Natural Step have been achieved, this is not still a sustainable activity.

Taking into account that principle number four posed by The Natural Step states that in a sustainable society, there should not be structural barriers that affect the health of people, their influence, competence, and impartiality and that if the contamination of sediments favors the bioavailability of mercury in people's diets, it is not possible to guarantee living conditions that allow the inhabitants of these municipalities to satisfy their needs and, therefore, the fourth principle of sustainability of The Natural Step cannot be achieved. Thus, if work is not continued to achieve these aspects, it will not be possible to comply with the principles of sustainability.

Regarding the third principle of The Natural Step, there is no data on the physical degradation resulting from small artisanal gold mining in Northeast Antioquia. Consequently, this is an aspect that needs special attention if it is to contribute to the construction of the sustainability of this activity.

### 4.2. Economic Sustainability

The use of whole ore amalgamation can be analyzed in two ways: the first has to do with the ratio of the use of mercury. The high range that has been reported implies a greater investment in the transformation process to obtain a poor amount of gold. Thus, the maximum potential of the mineral is lost and, consequently, a potentially higher profit is lost as well. In this way, it is very difficult to comply with the second principle of sustainability in mining proposed by Eggert [26], which consists of ensuring that mineral development is socially and economically efficient. In this case, the economic benefits are not maximized, and the expenses and possible income are not clear.

The second way is to focus the analysis on the word "efficient." With 30% gold recovery, it cannot be said that there is an efficient process. If the 50% of gold recovery proposed by Veiga [17] is preserved, it could be said that it is a process that recovers half of the gold present in the mineral and that the other half is discarded and would therefore generate half of the possible profits. As such, in this case, not only is there an inefficient mineral development, but the possible profits that can be obtained from the operation are not maximized, which violates the same principle of sustainability in mining mentioned in the previous paragraph.

On the other hand, the panorama has been changing since Veiga's [17] reports, and it can be said that in this aspect (in Northeast Antioquia), there has also been work on sustainability without talking about sustainability and without having a plan. Precisely, when

the University of British Columbia trained and empowered the owners of the processing plants (or "entables") at the Portovelo plant in Ecuador, about 39 of them changed their processing model to the one studied in the visit [18].

Due to a lack of capital, many other businesses were unable to completely transform their model but adopted pre-amalgamation concentration methods. Therefore, with the application of these changes, not only was whole ore amalgamation stopped, but the efficiency of gold extraction also increased, profits could be maximized a little more, and it could contribute to stopping the non-compliance of principle number two proposed by Eggert [26].

It is important to note that, given the lack of an official census for artisanal mining in Northeast Antioquia, it is very difficult to know how many operations and how many projects have yet to change their gold transformation methods. Therefore, if work is not continued on in this aspect, artisanal and small-scale gold mining in Northeast Antioquia will not be sustainable.

### 4.3. Sociocultural Sustainability

As already observed in the previous section, concentration processes highly contribute to the sustainability of small artisanal gold mining in Northeast Antioquia. However, if a culture is imparted in this region that allows changing the quick and easy method for the method that produces a greater added value to continue, it will be possible to contribute even more to the sustainability of small artisanal gold mining.

Thus, in the same way that retorts and concentrator equipment have become part of the gold transformation culture, mercury-free processes can as well. In this way, not only does it contribute to the environmental scope of sustainability, but the percentage of gold recovery is also maximized and thus complies with principle number two of sustainability in mining, which consists of maximizing the benefits of the project.

Until now, how the situation in Northeast Antioquia can affect principles 3 and 4 of sustainability in mining proposed by Eggert [26] has not been addressed at the sociocultural level. This is mainly because the type of culture that still predominates (subsistence) today in this region does not allow for these to principles to be adequately addressed. This is confirmed by the MMSD [24], which argues that there are several difficulties in achieving the ideal of a fair and equitable distribution of costs, benefits, and risks arising from mining activity:

- The first of the difficulties lies in how to identify and evaluate these costs, benefits, and risks, considering the cultural, economic, and environmental attributes [24].
- The second is more focused on the culture, policies, and laws of companies—which may well be an artisanal gold mining project—which on many occasions, hinder the transparency with which this issue is addressed [24].

That said, does a subsistence culture allow the generation of surpluses as a result of mining development? Additionally, what is the fairway to distribute the surpluses identified in this case?

These principles could be approached with fewer obstacles once the culture of gold recovery changes since a greater profit (product of a greater recovery of gold) could generate surpluses.

### 5. Conclusions

Based on the first sustainability perspective presented by Giovannoni and Fabietti [2], it was possible to identify that sustainability is a very broad term, which has different proposals, principles, and approaches depending on the application scenario. Thus, with the introduction of positions by different authors such as Porritt and Eggert, it was possible to reaffirm the breadth of existing proposals. With the introduction of the different sustainability initiatives that have emerged over time, such as The Natural Step, emphasis has been placed on a variety of principles that the term sustainability implies. Finally, the Giovannoni and Fabietti definition of business sustainability shows that there are different approaches to sustainability depending on the sector.

Similarly, with the presentation of these different definitions and positions, it was possible to establish that sustainability emerged within the field of renewable resources; therefore, it was necessary to identify whether it was possible to have a sustainability approach for an activity such as mining related to the extraction of non-renewable resources and which is the sector in which this research is concentrated. Eggert's definition of sustainability in mining demonstrated that artisanal and small-scale gold mining can have an approach towards sustainability, and with the presentation of its four principles, it was also demonstrated that this type of mining can maintain this approach's principles of sustainability even though it centers non-renewable resources. In addition, through the MMSD project and its seven questions, a tool for the evaluation and planning of sustainability in artisanal and small-scale gold mining was obtained.

Second, the presentation of the situation of small artisanal gold mining in Northeast Antioquia was a key aspect of this research because it made it possible to link theory and reality. With the exposure of mercury problems that ranked the Department of Antioquia as the most polluted by this metal in the world in 2010, the subsequent reductions in mercury emissions, the introduction of clean technologies in the gold transformation processes, and the training of personnel by the Initiatives of the University of British Columbia and various initiatives led by the Universidad Nacional de Colombia, it was possible to link numerous actions and activities conducted in reality with the theories proposed for sustainability in mining. From this link, it was found that these activities have contributed to the sustainability of small artisanal gold mining in Northeast Antioquia and that it is possible to organize a plan based on theories and principles of sustainability in mining.

Third, through the exposition of sustainability theories, three common factors could be identified within the various positions raised—the environmental, economic, and socio-cultural dimensions. With these, different needs were recognized in small artisanal gold mining in Northeast Antioquia that still require action.

**Author Contributions:** Conceptualization: Ó.J.R.B. and L.E.M.M.; Methodology: L.E.M.M.; Validation: Ó.J.R.B.; Formal analysis: L.E.M.M.; Investigation: L.E.M.M.; Data curation: Ó.J.R.B.; Writing—original draft preparation: L.E.M.M.; Writing—review and editing: Ó.J.R.B.; Visualization: L.E.M.M.; Supervision: Ó.J.R.B. All authors have read and agreed to the published version of the manuscript.

**Funding:** This research received no external funding.

**Institutional Review Board Statement:** Not applicable.

**Informed Consent Statement:** Not applicable.

**Data Availability Statement:** Not applicable.

**Conflicts of Interest:** The authors declare no conflict of interest.

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
