# Peer review of "Sustainability of the Artisanal and Small-Scale Gold Mining in Northeast Antioquia-Colombia"

_sustainability, doi:10.3390/su13169345_

Round 1

Reviewer 1 Report

This work is very interesting and illustrates how the theory of sustainability in mining meets reality in Antioquia-Colombia. A thorough review and explanation of sustainability of ASM gold mining.

I am missing several references on the text, which can be found in the reference list. Please put them into the text as well, wherever applicable.

minor grammatical and visual mistakes that need to be noticed and corrected. For instance, some table contents and figures need improvement.

I am missing a comment on the unregulated nature of ASM mining and how this affects sustainability. For your further understanding please see my comments on the manuscript (page 2).

Overall, it is very detailed work.

Author Response

Thank you very much for all the comments.

We have taken all the recommendations into account and have included them in the text. With respect to some of the comments, they can be supplemented by the following:

I am missing a comment on the unregulated nature of ASM mining and how this affects sustainability. For your further understanding please see my comments on the manuscript (page 2).

This has been included in the text

What I am missing from this work is addressing the issue of the unregulated depletion of reserves that usually comes with ASM. ASM producers tend to exploit the high (and medium) ore grade parts of a deposit and leave the lower grade blocks, which, however could be economic if conventional large-scale mining was to be applied. I encourage the authors to discuss this important issue as well.

We have discussed it in the document, although going deeper into these issues would imply changing the general direction of the text. However, the discussion has been expanded

Is there not a more up-to-date list of mines? 2011 is a decade ago

In a country like Colombia this information is very sensitive and is not updated for security reasons. There is no information available and this list has been placed for reference, but it is difficult to get something more current.

Reviewer 2 Report

The article is describing the situation of small-scale gold mines in the northeast Antioguia region in Columbia in relation to the concept of sustainability. It contains a description of sustainability concepts and the present situation in NE Antioquia regarding mercury emissions related to gold mining.

The article is very descriptive and doesn’t contain any information that hasn’t been published elsewhere before. This article only replicates concepts from other authors but doesn’t define any new sustainability concept like stated in the goal of the study (“…, this research aims to define the concepts of sustainability and relate them with small artisanal gold mining”). The current situation in Antioquia is more or less aligned with the requirements of Eggert 2019 or Natural Step. To draw the conclusion in the end that the environmental, economic and social dimension of sustainability are identified is rather weak. The article urgently needs language use review – sometimes strange terms are used, that wouldn’t be used by native speakers.

In the following some detailed comments:

14: „…in the Northeast Antioguia…” : include region or skip “the”

16-17: what does it mean “which are the environmental, economic…. dimensions” is this related to the “three common factors” or to “the various positions…”? The wording of the phrase should be changed to make it more clear, what is meant here.

22: “breadth”? Phrase, English use should be improved: e.g. change to: ”Owing to the broad scope of the term,…. “the dynamic…”

49 “water bodies” instead of “bodies of water”

phrase 50-51 need language revise – this is not a proper use of English

66: highest contamination by mercury in water or in human bodies? Could be specified a little more

74 could you express the gold production amount in kg or tones? Would be more convenient to read

94-98 Probably a little bit more could be explained, what with the results from the literature review is done then. How it is structured or processed. e.g. will there be categories formed or how important issues will be distinguished from unimportant etc. All this would be necessary to mention in the methodology part

99 (2.1) The description of the sustainability concepts is in my view not a part of the applied methodology. Because the methodology is applied to align concepts with the evaluation object (artisanal mining). This applies also to al the other sub-chapters presently located in chapter 2 – they are not describing the methodology. they are the main part of the study

107: a year is missing afte “Porritt” (I gues it is a citation?

  1. the citation in text is strange (display just author and year

116: Reference for Gruber 1989 is missing in the list of references

120-121: the citation in text is strange (display just author and year). And, just one colon after the citation

138-142: This was already stated in the introduction. Doesn’t have to be stated here again word by word. Moreover this reference is quite old, especially when looking on the dynamic in this field (as you also mentioned in the introduction). Perceptions might have changed meanwhile.

204: Probably the investigated region could be described in the beginning of the chapter and then the information about sustainability concepts

227-228: highest mercury contamination in the soil? or which environmental compartment?

273-276 Mercury concentration where? which environmental compartment is meant?

338-339 (and general): the previous text was just a description of the status quo and a literature review. There are just statements how it should be, but nothing about how these requirements could be transferred into practice.

341-343: This is common sense – the three dimensions of sustainability are the starting point of the assessment and not it result. Of course the three dimensions of sustainability can be identified from a scientific work dealing with sustainability – this is not a result?!

344 blank is missing (into3)

356-378: each time “Natural Step” is mentioned a reference is missing

446-448: this is not an original finding neither of this article nor of the article from Giovannoni & fabietti – this is common sense and widely known. The interesting part would be, how this braoad term can be made operable for the artisanal mining industry – but this was not elaborated

461-465 leaves the question behind, what the original work of this article is, if already Eggerts and the MMSD project demonstrating how the sustainability concept can be applied to mining

479-481 as already commented in line 341-343. This is not a result or a finding of this study

Author Response

Thank you very much for all the comments.

We have taken all the recommendations into account and have included them in the text. With respect to some of the comments, they can be supplemented by the following:

The article is very descriptive and doesn’t contain any information that hasn’t been published elsewhere before. This article only replicates concepts from other authors but doesn’t define any new sustainability concept like stated in the goal of the study (“…, this research aims to define the concepts of sustainability and relate them with small artisanal gold mining”).

Reference is made to the Colombian case, which is very particular due to the country's own conditions of violence, which hinders and delays the application of sustainability concepts and that is the novelty of the article. Colombia is a very complex case, because in addition to security problems there are guerrillas, military, drug trafficking and common crime, which makes it very different. All these issues have been addressed in the development of the corrected text. 

338-339 (and general): the previous text was just a description of the status quo and a literature review. There are just statements how it should be, but nothing about how these requirements could be transferred into practice.

Thank you for your comment. This has been taken into account and implemented in the new version of the article attached.

341-343: This is common sense – the three dimensions of sustainability are the starting point of the assessment and not it result. Of course the three dimensions of sustainability can be identified from a scientific work dealing with sustainability – this is not a result?!

Thank you for your comment. This has been taken into account and implemented in the new version of the article attached.

446-448: this is not an original finding neither of this article nor of the article from Giovannoni & fabietti – this is common sense and widely known. The interesting part would be, how this braoad term can be made operable for the artisanal mining industry – but this was not elaborated

Thank you for your comment. This has been taken into account and implemented in the new version of the article attached.

461-465 leaves the question behind, what the original work of this article is, if already Eggerts and the MMSD project demonstrating how the sustainability concept can be applied to mining.

The originality of the article is based on the particular conditions of the Colombian case. As explained in the previous question.

479-481 as already commented in line 341-343. This is not a result or a finding of this study.

Thank you for your comment. This has been taken into account and implemented in the new version of the article attached.

Round 2

Reviewer 2 Report

-